# Compliance to Multidisciplinary Lifestyle Intervention Decreases Blood Pressure in Patients with Resistant Hypertension: A Cross-Sectional Pilot Study

**DOI:** 10.3390/jcm12020679

**Published:** 2023-01-15

**Authors:** Eugenia Espinel, María Antonia Azancot, Alba Gomez, Anna Beneria, Anna Caraben, Laura Andurell, Pilar Delgado, Helena Castañé, Jorge Joven, Daniel Seron

**Affiliations:** 1Department of Nephrology, Hospital Universitari de la Vall d’Hebrón, Universitat Autònoma de Barcelona, 08035 Barcelona, Spain; 2Department of Physical Medicine and Rehabilitation, Hospital Universitari de la Vall d’Hebrón, Universitat Autònoma de Barcelona, 08035 Barcelona, Spain; 3Department of Psychiatry, Hospital Universitari de la Vall d’Hebrón, Universitat Autònoma de Barcelona, 08035 Barcelona, Spain; 4Department of Endocrinology and Nutrition, Hospital Universitari de la Vall d’Hebrón, Universitat Autònoma de Barcelona, 08035 Barcelona, Spain; 5Department of Neurology, Hospital Universitari de la Vall d’Hebrón, Universitat Autònoma de Barcelona, 08035 Barcelona, Spain; 6Unitat de Recerca Biomèdica, Hospital Universitari de Sant Joan, Institut d’Investigació Sanitària Pere Virgili, Universitat Rovira i Virgili, 43201 Reus, Spain

**Keywords:** hypertension, therapeutics, weight management

## Abstract

Hypertension is a common chronic medical condition. Treatment is not satisfactory in a significant proportion of patients with primary hypertension, despite the concurrent use of three or more medications with different mechanisms of action. Such treatment-resistant hypertension is a clinical challenge associated with poor prognosis and needs further investigation. The efficacy of lifestyle changes has not been established yet in patients with resistant hypertension, and educational efforts appear clinically irrelevant in patients who must achieve behavioral changes without supervision. A 6-month multidisciplinary pilot intervention enrolled 50 patients with established resistant hypertension. The aims were: (1) to examine whether intensive and supervised lifestyle changes contribute to decreasing blood pressure in this condition, and (2) to identify which components affect compliance and feasibility. The program provided intensive changes in nutrition, physical exercise, and control of sleep disturbances supervised by nutritionists, physiotherapists, and psychologists. Nurses and pharmacists followed up on adherence to the antihypertensive medication. The primary outcome was 24 h blood pressure control. Data in patients with full compliance (n = 30) indicate that lifestyle modifications in resistant hypertension significantly reduced 24 h both systolic and diastolic blood pressure (*p* < 0.01), body mass index (*p* < 0.01), medication burden (*p* = 0.04), improving physical fitness, and cardiovascular risk markers such as heart rate (*p* = 0.01) and augmentation index (*p* = 0.02). The adherence to the intervention was moderate, with an attrition rate of 12%. A modified version reducing visits and explorations will likely improve compliance and can be used to assess the long-term maintenance of these benefits in managing resistant hypertension by diverse healthcare providers.

## 1. Introduction

Despite improvements in hypertension awareness and treatment, the associated risk of death is rising because target organ damage remains when blood pressure (BP) goals are not achieved [1]. Clinically, addressing common problems such as avoiding substance abuse and medications that raise BP, inaccurate BP measurement, nonadherence to antihypertensive agents, and misdiagnosis of reversible causes of hypertension may improve BP control [2,3,4]. However, managing patients with treatment-resistant hypertension (RHT), i.e., ambulatory BP not at goal despite prescribing three or more antihypertensive drugs, including diuretics, alpha, and beta-blockers at optimal and best-tolerated dosages, is complex and faces significant challenges [5,6,7]. Because epidemiologic studies usually do not incorporate all the information regarding such a definition, prevalence is difficult to calculate but estimated as >10% among individuals with primary hypertension [8], reaching more than 20% in some countries [8,9]. The concurrence of other clinical factors that may affect BP, namely sleep disturbances and obesity, is frequent and a consistent body of evidence supports the notion that RHT is associated with a significantly increased all-cause mortality and morbidity [2,5,6,8,10]. Consequently, uncontrolled forms of hypertension are the cause of considerable concern among clinicians suggesting the existence of mechanisms underlying RHT that need to be clarified and incorporated into therapeutics [11,12].

In defiance of guidelines for pharmacological treatment and assessment of drug combinations, multi-drug treatment is not as effective as desirable [2,13,14]. Approved therapies cover multiple pathological pathways, including vasodilation, regulation of salt and water, antagonism or blockade of the renin-angiotensin-aldosterone system, regulation of intracellular calcium, and sympatholytic activity [15]. Because the hemodynamic effects of many of these drugs are unpredictable, there is room for novel therapeutic approaches [16]. The multifunctional peptide endothelin-1 provides an alternative pathway, and receptor antagonists are currently under investigation in managing RHT [17]. Likewise, because drugs with modest weight-reducing effects decrease BP in patients with hypertension [18], novel and highly effective agents for weight loss in patients with obesity may result in substantial improvement [19]. Invasive procedures are currently under assay in patients with RHT, but the long-term effectiveness and safety concerns of renal nerve ablation, baroreceptor stimulation, experimental devices, and bariatric surgery remain controversial [20,21,22,23].

Adopting a healthy cardiovascular lifestyle with increased exercise, dietary modification, and weight loss is often recommended as the first step for treating high BP with remarkable efficacy in unmedicated or drug-controlled patients with hypertension [24,25,26,27]. Unfortunately, no trials robustly establish the effectiveness of lifestyle modifications, and nonpharmacological management in patients with RHT remains underestimated [14]. A recent study [28] demonstrated that care in a cardiac rehabilitation setting is more effective than simple counseling without further assistance. Indeed, educational efforts are considered too time-consuming and appear clinically irrelevant in patients who have to achieve behavioral changes on their own [29]. Because the pathophysiology of RHT is unique, the effectiveness of lifestyle changes can be neither assumed nor neglected, considering the significant effects recently reported in the stroke-prone population [30]. To add further insight into the effectiveness of supervised care in maximizing lifestyle changes in patients with long-established RHT and high cardiovascular risk, we have assessed 24 h BP reductions during a 6-month intensive, multidisciplinary intervention. Our pilot data suggest feasible designs for future studies exploring long-term benefits.

## 2. Materials and Methods

### 2.1. Pilot Study: Overview

Behavioral interventions need decisions about their feasibility to select and adapt appropriate methods rather than being derived from highly controlled efficacy studies. We designed a prospective cross-sectional pilot study in patients with RHT, a population target with unique particularities, through a 6-month program providing comprehensive care. The organization of the study included a multidisciplinary team of nurses, pharmacists, nutritionists, physiotherapists, psychologists, and physicians with the necessary skills. Those involved in implementing the program performed monthly visits and procedures to compare baseline and post-intervention BP measurements. Personalized sessions emphasized healthy lifestyle changes, including exercise programs and nutritional counseling. The implications of how the intervention would fit with daily-life activities, perceived sustainability, and costs within organizational goals focused on our concerns. The organization and procedures were approved by the Institutional Review Board [PR(AG)207/2015] and were monitored by the local Ethics Committee of the Vall d’Hebron Hospital. All procedures were conducted in accordance with the Good Clinical Practice Guidelines of the Health Department of Generalitat de Catalunya and were in compliance with the Declaration of Helsinki. All participants provided written informed consent. Details will be made available upon reasonable request. Rationale about feasibility and gains in the precision and the mean and variance of BP measurements indicated the justifications for a minimum sample size of 24 cases [31].

### 2.2. Participants

Those with secondary causes of hypertension or clinically severe diabetes and/or ischemic heart disease were considered ineligible among contacted patients. Other initial exclusion criteria included the use of illicit drugs, including ethanol in excess, and previous prescription of weight-loss pills and medications that can raise BP. Fifty patients with true RHT were enrolled among those regularly attending our clinic with BP values > 130/80 mmHg when three or more different classes of antihypertensive drugs, including a diuretic at optimized dosages, were prescribed or the need for four or more drugs to achieve BP < 130/80 mmHg. As RHT was a diagnosis of exclusion, ambulatory BP monitoring was not only an essential component in the evaluation but also reinforced lifestyle modifications. Self-reports and reports from attending nurses and pharmacists confirmed excellent adherence to antihypertensives.

### 2.3. Clinical Assessment

Recruited patients with RHT were reevaluated at baseline to ensure the diagnosis, and the lead clinician reviewed medications and supervised clinical adherence according to guidelines [2]. All enrolled patients had a series of laboratory variables and 24 h ambulatory BP measurements on study entry and after the intervention, including circadian patterns, BP variability, and morning BP surge, according to the European Society of Hypertension guidelines [32,33,34]. These measurements were the primary outcome. To ensure that patients received the appropriate sleep, we measured the amount of uninterrupted sleep time and used polysomnography to confirm the presence of obstructive sleep apnea. All patients diagnosed with the condition received guidance from their doctor to maximize the effect of continuous positive airway pressure therapy and to assess correct implementation. On a monthly basis, we conducted personalized nutritional education sessions with the aid of nurses and dietitians to promote and reinforce the adoption of a healthy and low-salt diet. Our specialists provided a customized and hypocaloric diet that looks to account for each patient’s individual preferences and aversions, tailored to their specific clinical situations. A physiotherapist provided instructions to patients on how to improve their physical strength and oversee lifestyle changes through a personalized exercise program. The patients were encouraged to engage in daily walks lasting between 45 and 60 min to promote physical activity. Psychologists conducted evaluations to identify any relevant psychosocial factors that might be impacting the patients, and also assessed the potential for cognitive decline to customize their counseling on nutrition and exercise to the individual needs. They evaluated the patients’ adherence to the prescribed treatment plan. We also explored changes from before to after intervention in cardiovascular biomarkers. Brain magnetic resonance imaging was used to explore cerebrovascular disease. Echocardiography was used to assess cardiac geometry and left ventricular function. Endothelial function, evaluated with brachial artery flow-mediated vasodilation, carotid intima-media thickness, and carotid-femoral pulse wave velocity, were considered surrogates of arterial stiffness. All appointments were coordinated to ensure the monthly visits to the Clinic.

### 2.4. Statistics

We only included data from patients with full compliance and no missing data. Numbers and percentages expressed qualitative variables, and mean, and standard deviation described quantitative variables. Therefore, we used a paired *t*-test, chi-square test, or McNemar test, and analyses of differences conducted with SPSS 24 (IBM, Chicago, IL, USA) were significant when *p*-values were <0.05. Plots from GraphPad Prism 6.01 (GraphPad Software, San Diego, CA, USA) illustrated changes in BP measurements. Because the response was not uniform, we also used R (version 4.0.2) to calculate the Pearson pairwise correlation matrices, MetaboAnalystR [35] to run Partial Least Square—Discriminant Analysis (PLS-DA), and to assess predictive values of variables with the most discriminant capacity. The scikit-learn package [36] in Python (version 3.8.12) explored multiple linear regression and random forest models.

## 3. Results

### 3.1. Adherence to Intervention

Patients were initially contacted (n = 62) among those regularly attending our clinic with strict diagnostic criteria for RHT and without symptomatic conditions or adverse effects from drug dosages. In the first interview, patients answered about their perception of the importance of the intervention, social/physical factors, commitment, and ability to match schedules for six months. Fifty patients participated in addressing the importance of compliance with lifestyle changes. There was a 12% attrition rate, and 28% of patients missed >1 visit. Accordingly, we retained data from 30 patients that completed the intervention for final analyses and considerations on the appropriateness of intervention and recruitment strategies (Appendix A).

### 3.2. Clinical Characteristics and Laboratory Testing

Table 1 depicts the selected variables. Participants were, on average, 60 years old, with modest differences between men and women. They were ethnically homogeneous and sedentary with excess weight. There were patients with obesity (Body mass index (BMI) > 30 kg/m^2^). Consistent with the view that RHT is associated with a poor prognosis, clinical examination and records confirmed high cardiovascular risk, one-third had diabetes, and 63% had metabolic syndrome. Patients have never smoked or quit cigarettes for more than 20 months. A significant proportion of patients (n = 20 (71%)) had at least one marker for cerebral small vessel disease with cognitive levels between 21 and 28 according to an MoCA score adjusted for age and education level, which did not vary after the intervention. Laboratory testing (Appendix A) was planned to assist in cardiovascular risk assessment and detect electrolyte changes. Changes after intervention were considered negligible but revealed that the urinary albumin-to-creatinine ratio was responsive to treatment and that diet and lifestyle are associated with decreased serum C-reactive protein concentrations [37].

### 3.3. Blood Pressure Outcomes

Clinic systolic (SBP) and diastolic (DBP) BP measures were performed during each visit and indicated a substantial but variable improvement. To assess the effect of the intervention, we compared the readings provided by the 24 h ambulatory blood pressure monitoring system during baseline and post-intervention. Individual changes demonstrated improvement, and most patients achieved the desired goal (Figure 1a). There were also modest but significant post-intervention changes in BP antihypertensive medications both in number and daily-defined dose (Figure 1b,c), indicating successful variations in prescriptions based on clinical factors. However, the response to intervention in the primary outcome was not homogeneous (Figure 1d). Individual changes identified those patients arbitrarily classified according to response in the primary outcome, and a significant number of patients (20%) were considered non-responders (Appendix A). We did not find objective reasons for this lack of response. However, the exploratory data analysis heat map showed a statistically non-significant trend between baseline BP and improvement, where a better response was associated with worse baseline control (Appendix A). Among patients’ independent-BP characteristics, multivariate analyses and random forest identified predictors such as the serum concentration of renin and C-reactive protein, and protein and albumin excretion in urine, as the variables with the most discriminant ability between patients with a different response. This may indicate the importance of inflammatory status and hemodynamic changes caused by prescribed drugs (Appendix A).

Table 2 presents mean values in baseline pre- and post-intervention 24 h ambulatory blood pressure. There was a substantial mean reduction in 24 h ambulatory BP (−14.0/−8.5 mmHg (SBP/DBP)), and all BP-associated variables decreased without apparent changes in peripheral vascular resistance.

Lifestyle changes were not associated with managing BP variability, but we found more non-dippers and fewer dippers after the intervention. Our measurements compare BP variations over 24 h and the effect likely associated with the intervention. We found a modest but significant reduction in DBP variability. However, there were no significant differences in changes typically related to biological rhythms measured as dispersion or weighted standard deviation or in the sequence of measurements over the six months assessed with average real variability.

### 3.4. Cardiovascular Risk Stratification

Few studies reported the independent contribution of BP variability and nocturnal dipping to cardiovascular risk stratification. However, there is a temporal relationship between morning blood pressure surge (MBPS) and the peak incidence of cardiovascular events. After the intervention, MBPS size decreased, measured as sleep-through and pre-waking MBPS (Table 2). We also found a potential improvement in endothelial dysfunction, but the variations in arterial diameter using reactive hyperemia (2.7 ± 3.6% at baseline and 5.4 ± 6.9% after intervention) did not reach statistical significance. As significant associations were higher at baseline than after intervention, we concluded that changes in BP and all measured variables during the intervention positively affected the cardiovascular risk (Figure 2a,b). Risk-assessment methods commonly used in our geographical region also depicted beneficial effects from the intervention and a significant decrease in heart rate (Figure 2c,d). Moreover, there were substantial decreases in surrogates of arterial stiffness after the intervention (Figure 2e).

### 3.5. The Importance of Weight Loss

The relationships between lifestyle interventions, weight loss, and increased physical activity were considered relevant factors. All patients had at baseline poor dietary habits and a sedentary lifestyle. They were considered “unfit” or low “fit”. After the intervention, patients walked routinely for more than 60 min daily. Resistance training was progressively accepted throughout the subsequent visits, and portable devices assisted in evaluating sleep patterns and documenting greater activity measured as average steps per week. Although we did not objectively examine changes in aerobic fitness, all patients moved to a clinically higher fitness category. Self-reported satisfaction with dietary advice was excellent, although adherence to a salt-free diet was deemed suboptimal in some patients with mean reductions in sodium intake of 290 mg/day [95% CI, −50 to −482] according to the food frequency questionnaire. The mean decrease in calorie intake from baseline to post-intervention was modest, −256 kcal/day [95% CI, −98 to −382], but associated with relatively increased servings per day of vegetables, fruits, nuts, and legumes. Participants exhibited substantial weight loss with significant changes in BMI that were not appreciated by measurements of waist circumference (Figure 3a,b). Of note, we observed at baseline that BMI correlated with nighttime BP and 24 h DBP, both central and peripheral, and such correlation was no longer significant after the intervention (Figure 3c,d).

## 4. Discussion

Patients with RHT, who do not respond well to multiple antihypertensive medications, need unique consideration. Mortality, the prevalence of end-organ damage, weight excess, and cardiovascular risk are higher than in effortlessly controlled hypertensive patients [38]. There is a major medical need for additional therapy targeting the etiology and associated diseases that can be combined with existing therapies.

It is relevant to identify effective behavioral changes for lowering BP, but the impact of lifestyle changes on this population is unclear. The few available data are promising and demonstrate that programs for patients that have to achieve behavioral changes on their own perform poorly [28,39,40]. However, patients with RHT obtained significant BP reductions with a lower medication burden within this 6-month organizational change. BP reductions of this magnitude, a significant decrease in heart rate, and beneficial variations in arterial stiffness are likely associated with positive outcomes [41,42,43]. Similarly, controlling the time of uninterrupted sleep, the substantial reduction in morning BP surge, improved fitness, and weight loss may also translate to improved cardiovascular risk [44,45,46].

Weight loss has been demonstrated to play a significant role in managing resistant hypertension [47,48]. Obesity is a common risk factor for the development of hypertension, and weight loss can help to reduce blood pressure levels in patients with obesity-related hypertension. In some cases, even a modest weight loss of 5–10% of body weight can significantly improve blood pressure. In addition to its effects on blood pressure, weight loss can also have other cardiovascular benefits, such as reducing the risk of diabetes, coronary artery disease, and stroke [23]. Weight loss that occurs with lifestyle modifications has been demonstrated to minimize hypertensive patients’ reliance on antihypertensive medications or potentially eliminate their need [26].

Our practice-based evidence confirmed the efficacy of behavioral interventions and provided numerically larger BP reductions than those observed with invasive procedures [20,23,49,50]. However, interventions in health promotion must include changeable behaviors to include those key factors with the greatest likelihood of being efficacious. Our feasibility study determined that the intervention is appropriate for further testing and identified what modifications may improve success in evaluating effectiveness in the real world. Acceptability and demand from participants and those involved in implementing the program were high, but in the post-study analysis, we questioned selected activities. For instance, the requisite for full compliance must accommodate the population’s requirements. The implementation has been repurposed to increase practicality and expansion to community care settings through group sessions followed by online assistance and follow-up. Similarly, there is perceived appropriateness of the continuous use of portable devices and remote control in the exercise program. With perceived sustainability, a modified format with improved generalizability and lower costs to the organization is currently ongoing. The ultimate challenge is to assess long-term effectiveness and sustainability.

As this pilot study was planned as a 6-month intervention, the data on persistence and long-term effectiveness is lacking, which represents a limitation. In addition, the sample size should be increased to accommodate more and all types of patients (i.e., smokers, as they represent a significant proportion) and to assess the heterogeneity that we have observed in response to treatment.

To conclude, maximizing lifestyle intervention is a beneficial therapeutic strategy in patients with RHT with the ability to achieve significant BP reductions. Interventions, likely associated with weight loss and fitness, yield positive outcomes and should not be disregarded in patients with poor fitness, weight excess, and major cardiovascular damage. The success of these interventions depends not only on the patient’s consent and motivation but also on their willingness to attend visits with the multidisciplinary team and adhere to the program. This requires strong coordination and communication among the members of the multidisciplinary team in order to ensure that the program is feasible and sustainable for the patient. It is crucial for the patient to receive ongoing support and guidance in order to maintain long-term healthful behaviors.

## Figures and Tables

**Figure 1 jcm-12-00679-f001:**
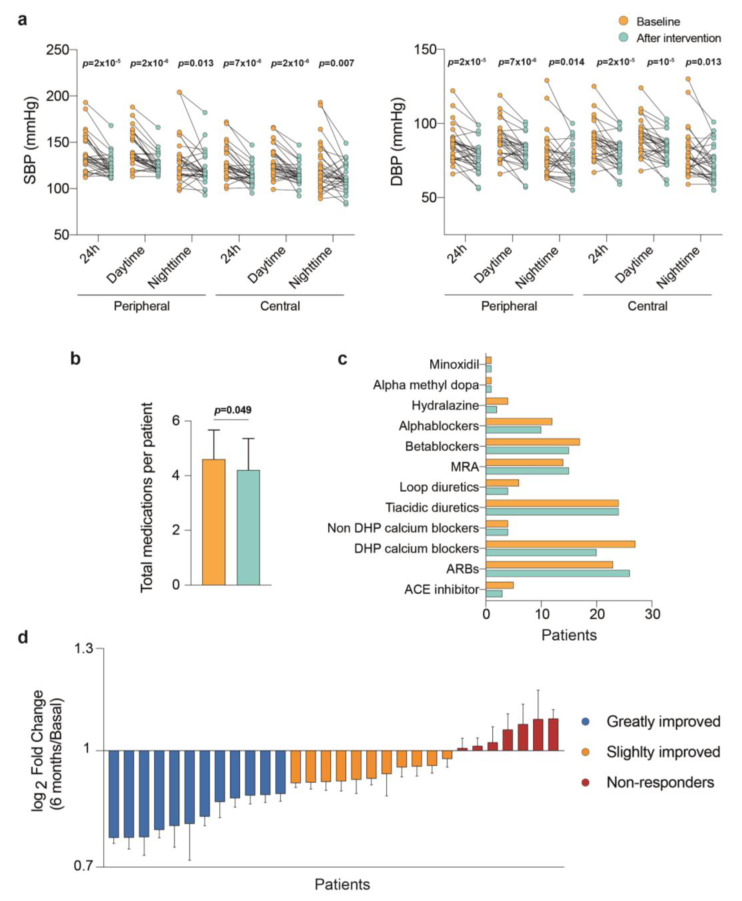
Lifestyle interventions decrease blood pressure in patients with resistant hypertension, but the response was not homogeneous. (**a**) Individual changes in systolic (SBP) and diastolic (DBP) blood pressure, (**b**,**c**) changes in medication burden from baseline to after intervention, (**d**) the response to intervention was not homogeneous.

**Figure 2 jcm-12-00679-f002:**
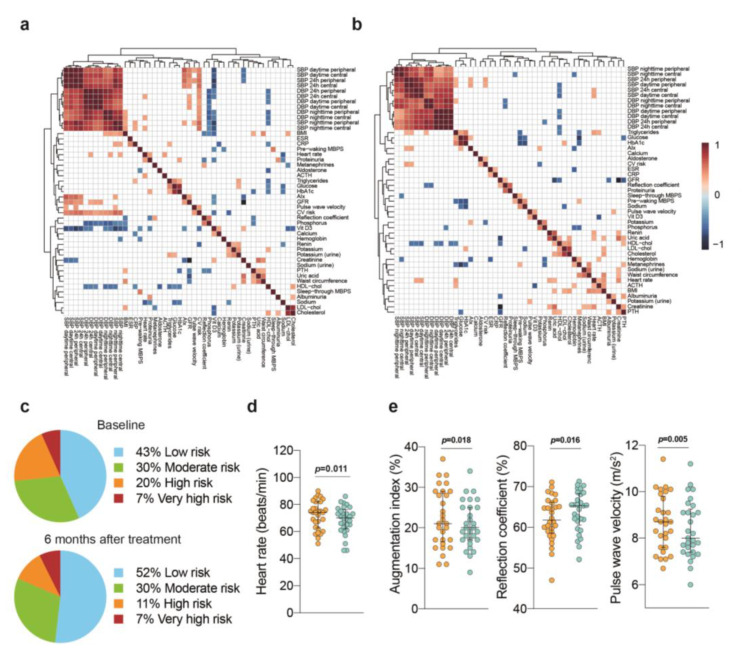
Potential relevance in cardiovascular risk. (**a**) Correlation matrix indicating significant associations between BP and variables associated with cardiovascular risk, which disappeared after the intervention (**b**). Consistently, assessment methods showed decreased risk (**c**). Heart rate (**d**) and surrogates for arterial stiffness (**e**) also diminished after the intervention.

**Figure 3 jcm-12-00679-f003:**
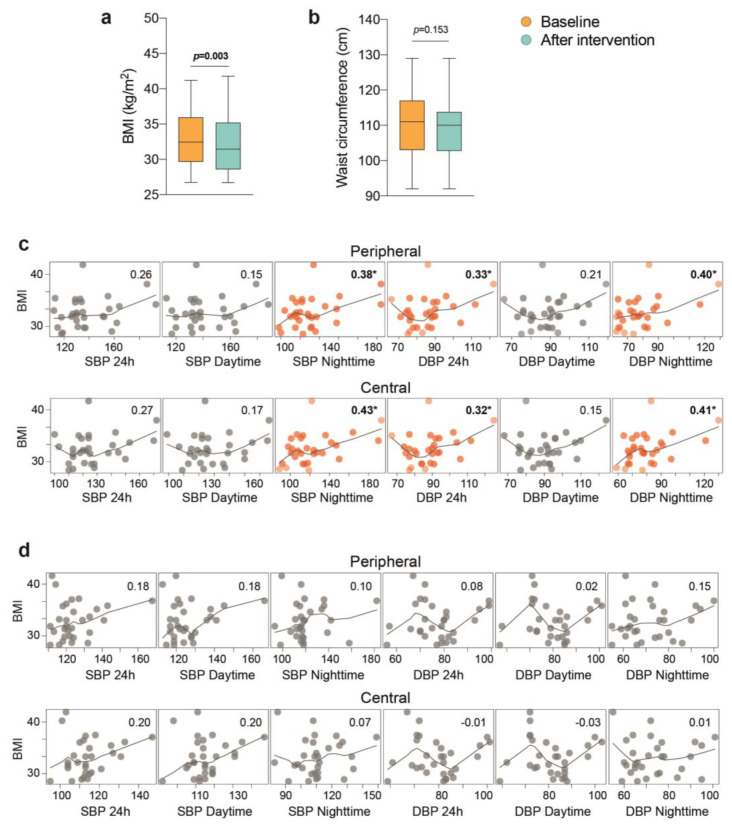
The intervention modified the relationship between weight loss and blood pressure in patients with resistant hypertension. (**a**) Body mass index (BMI). (**b**) Waist circumference. Significant correlation plots with the fitted line at baseline (**c**) of systolic (SBP) and (**d**) diastolic (DBP) blood pressure with BMI were not maintained after the intervention. Pearson’s correlation coefficient is represented in the upper and right part of each plot, and significance is with an asterisk and numbers in bold.

**Table 1 jcm-12-00679-t001:** Selected characteristics in the study group.

	Patients with RHT (n = 30)
Age (years)	59.6 ± 8.8
Sex, men, n (%)	22 (66)
BMI (kg/m^2^)	33.1 ± 4.4
Waist circumference (m)	1.1 ± 0.1
24 h SBP (mmHg)	138.3 ± 20.1
24 h DBP (mmHg)	85.7 ± 12.1
T2DM, n (%)	12 (36)
Dyslipidemia, n (%)	23 (76)
OSAS, n (%)	18 (60)
Metabolic syndrome, n (%)	19 (63)
Atrial fibrillation, n (%)	1 (3)
Right branch bundle block, n (%)	4 (13)
Left branch bundle block, n (%)	2 (6)
Left ventricular hypertrophy, n (%)	27 (90)
Heart rate (beats/min)	72.2 ± 10.4
PWV > 10 m/seg, n (%)	7 (23)
Ankle-brachial index < 0.9, n (%)	1 (3)
Left atrial enlargement, n (%)	25 (83)
Carotid artery plaque, n (%)	23 (76)

BMI: body mass index; DBP: diastolic blood pressure; OSAS: obstructive sleep apnea syndrome; PWV: pulse wave velocity; RHT: resistant hypertension; SBP: systolic blood pressure; T2DM: type 2 diabetes mellitus.

**Table 2 jcm-12-00679-t002:** Mean changes in ambulatory blood pressure and calculated variables.

	Baseline	After Intervention	*p*-Value
**Peripheral blood pressure**			
24 h SBP (mmHg)	138.3 ± 20.1	124.3 ± 11.8	**1.6 × 10^−5^**
24 h DBP (mmHg)	85.7 ± 12.1	77.2 ± 10.4	**2 × 10^−5^**
Daytime SBP (mmHg)	141.2 ± 18.8	126.1 ± 11	**2.5 × 10^−6^**
Daytime DBP (mmHg)	88.6 ± 11.6	79.7 ± 10.4	**7.3 × 10^−6^**
Nighttime SBP (mmHg)	131 ± 25.3	121.1 ± 17.7	**0.013**
Nighttime DBP (mmHg)	77.9 ± 15.3	72 ± 11.9	**0.014**
Heart rate (beats/min)	72.2 ± 10.4	69.3 ± 10.9	**0.038**
**Central hemodynamics**			
Central SBP 24 h (mmHg)	126.9 ± 17.7	114.1 ± 10.6	**6.7 × 10^−6^**
Central DBP 24 h (mmHg)	85.5 ± 12.3	79.5 ± 10.4	**1.8 × 10^−5^**
Central daytime SBP (mmHg)	128.4 ± 16.5	115.4 ± 10.8	**2.5 × 10^−6^**
Central nighttime DBP (mmHg)	90.2 ± 12.1	81.4 ± 10.3	**1.3 × 10^−6^**
Central daytime SBP (mmHg)	122.3 ± 25.8	110.9 ± 14.3	**0.007**
Central nighttime DBP (mmHg)	80.1 ± 15.7	73.4 ± 12.1	**0.013**
Cardiac output (L/min)	6.3 ± 7	6.3 ± 8.6	**0.013**
**Variability of arterial pressure**			
Systolic mean (mmHg)	132.5 ± 16	126.3 ± 14.1	0.064
Diastolic mean (mmHg)	82.5 ± 11.2	78.5 ± 10.9	**0.049**
Systolic SD (mmHg)	16.8 ± 4.9	15.1 ± 3.9	0.184
Diastolic SD (mmHg)	12.6 ± 3.1	11.7 ± 2.6	0.178
Systolic CV (mmHg)	12.6± 3.2	11.9 ± 2.5	0.229
Diastolic CV (mmHg)	15.4 ± 3.9	15.1 ± 3.3	0.685
Systolic ARV (mmHg)	14.5 ± 5.3	13.6 ± 5	0.375
Diastolic ARV (mmHg)	11.3 ± 3.6	10.3 ± 3.4	0.243
Weighted SD (mmHg)	22.3 ± 2	22.4 ± 1.5	0.948
**Surge morning**			
Sleep-through MBPS (mmHg)	26 ± 14.5	18.4 ± 14.9	0.061
Pre-waking MBPS (mmHg)	23.2 ± 20.5	12.4 ± 11.5	**0.036**

SBP: systolic blood pressure; DBP: diastolic blood pressure; MBPS: morning blood pressure surge; SD: standard deviation; CV: variation coefficient; ARV: average real variability.

## Data Availability

Request to access the data set and protocols from qualified researchers may be sent to the corresponding authors.

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
