# Peer review of "Compliance to Multidisciplinary Lifestyle Intervention Decreases Blood Pressure in Patients with Resistant Hypertension: A Cross-Sectional Pilot Study"

_jcm, 2023, doi:10.3390/jcm12020679_

Round 1
Reviewer 1 Report
Espinel E and coworkers evaluated the lifestyle changes in patients with resistant hypertension. The authors found that lifestyle modifications in resistant hypertension improved physical fitness and cardiovascular risk with significant reductions in blood pressure, body mass index, and medications.
This manuscript can give the role of a specific lifestyle program in resistant hypertension and provide a utility tool for treating patients with resistant hypertension.
I have a few points which deserve clarification.
The introduction is well-written. I suggest citing the following paper PMID: 22902872 DOI: 10.1097/HJH.0b013e3283577b20 ; PMID: 21677699 DOI: 10.1038/ajh.2011.106;
The authors should better specify the program's nutritional education sessions and exercise protocol.
Did the authors make a psychological program for sleep disturbance?
The discussion is poor. I suggest discussing the following point
- The role of the nutritional and exercise programs
- The role of results in weight loss and consequently in a more control of blood pressure
- The role of the intervention program and the need to reduce medications.
Author Response
"This manuscript can give the role of a specific lifestyle program in resistant hypertension and provide a utility tool for treating patients with resistant hypertension."
Response: Thank you for your overall positive appreciation and attention to the points deserving clarification.
Point 1: "The introduction is well-written. I suggest citing the following papers [...]."
Response: We considered those valuable references as contributions to be included in the text (line 54), and we performed the subsequent changes in numbering the references.
Points 2 and 3 combined: "The authors should better specify the program's nutritional education sessions and exercise protocol. Did the authors make a psychological program for sleep disturbance?"
Response: As suggested, we provide further details on the raised points to clarify the text, now rewritten in lines 149-164.
Point 4: "The discussion is poor. I suggest discussing the following points:
- The role of the nutritional and exercise programs
- The role of results in weight loss and consequently in a more control of blood pressure
- The role of the intervention program and the need to reduce medications."
Response: We have included these topics accordingly, supported by additional references (lines 322-330).
Thank you for your feedback.
Reviewer 2 Report
The prospective study proposed by Espinel et al. aimed to evaluate the efficacy and the feasibility of an intensive lifestyle modification program (by changing nutrition and promoting physical exercise, supervised by nutritionists, physiotherapists, and psychologists) lasting 6 months, in the treatment of resistant hypertension. The primary outcome was 24-hours blood pressure control. It is an interesting topic, but the sample size is too small, only 30 patients with resistant hypertension that completed the intervention. Moreover, the article requires significant revision, because it is not written in good scientific English.
Maior concerns:
· As mentioned above, the main problem is the low sample size, only 30 cases that completed the intervention for final analysis, out of about double of patients enrolled (50). This means that the intervention program is too intensive and difficult to follow, even in the presence of supervision and that with the end of the same, compliance will be even lower. · The time of follow-up is too short, only 6 months and there are no data on persistence of effectiveness on blood pressure control, months after the end of the intensive program. · Why did you select only not or formers smokers between enrolled patients? This represented a selection bias, in fact in real life, hypertensive and metabolic subjects, frequently are smokers. · In the main text were reported only data of efficacy of the intensive lifestyle modification program on parameters of ambulatory blood pressure monitoring and on surrogates of arterial stiffness and endothelial disfunction, by statistical univariata analysis. Only in the paragraph of the blood pressure outcomes was mentioned the execution of multivariate analysis, but without reporting the results. · I suggest reporting the results of multivariate statistical analysis. · In the conclusions the authors had not adequately reported the factors that allow the feasibility of the program, as to consent its application in real life, with the absence of supervision of the various specialists. Minor concerns: · I suggest correcting, in the introduction, the prevalence of resistance hypertension, because it is a condition more frequent than >10% of patients with arterial hypertension, as reported by the authors; · I suggest correcting the definition of resistant hypertension, as reported in the 2018 European Society of Hypertension Guidelines, with “office BP not at goal despite the prescription of three or more classes of antihypertensive medications, including a diuretic, and typically an ACE inhibitor or an Angiotensin receptor blocker and a calcium channel blocker at optimal or best tolerated dose”. · I suggest reporting in the main text, more in details, the practical modalities of execution of the intensive lifestyle modification program, that are only mentioned and are unclear. · The abstract is too conversational, I suggest mentioning the results of statistical univariate analysis on it.Author Response
"The prospective study proposed by Espinel et al. aimed to evaluate the efficacy and the feasibility of an intensive lifestyle modification program (by changing nutrition and promoting physical exercise, supervised by nutritionists, physiotherapists, and psychologists) lasting 6 months, in the treatment of resistant hypertension. The primary outcome was 24-hours blood pressure control. It is an interesting topic, but the sample size is too small, only 30 patients with resistant hypertension that completed the intervention. Moreover, the article requires significant revision, because it is not written in good scientific English."
Response: Thank you for your suggestions. Indeed, we are fully aware of the limitations of a pilot study. We addressed your comments throughout the text and the limitations expressly incorporated into the Materials and methods and Discussion sections.
Point 1: "As mentioned above, the main problem is the low sample size, only 30 cases that completed the intervention for final analysis,out of about double of patients enrolled (50). This means that the intervention program is too intensive and difficult to follow, even in the presence of supervision and that with the end of the same, compliance will be even lower."
Response: Your comment is correct. In a pilot study, these are results as well as limitations. Please note we have addressed this point in lines 120-122 and 348-351.
Point 2: "The time of follow-up is too short, only 6 months and there are no data on persistence of effectiveness on blood pressure control, months after the end of the intensive program."
Response: Your arguments are correct again. Being a pilot study, a 6-month follow-up was sufficient to draw our main conclusions (i.e., the need for modifications in this program). We have clarified this limitation in lines 347-348.
Point 3: "Why did you select only not or formers smokers between enrolled patients? This represented a selection bias, in fact in real life, hypertensive and metabolic subjects, frequently are smokers."
Response: We agree this represents a potential bias. The consequences in future decisions are clarified in lines 348-351.
Point 4: "In the main text were reported only data of efficacy of the intensive lifestyle modification program on parameters of ambulatory blood pressure monitoring and on surrogates of arterial stiffness and endothelial disfunction, by statistical univariata analysis. Only in the paragraph of the blood pressure outcomes was mentioned the execution of multivariate analysis, but without reporting the results. I suggest reporting the results of multivariate statistical analysis."
Response: There is probably a misunderstanding here. All multivariate analyses are reported. We have improved readability in lines 233-241.
Point 5: "In the conclusions the authors had not adequately reported the factors that allow the feasibility of the program, as to consent its application in real life, with the absence of supervision of the various specialists."
Response: We agree with these concerns and are now included in lines 357-362.
Point 6: "I suggest correcting, in the introduction, the prevalence of resistance hypertension, because it is a condition more frequent than >10% of patients with arterial hypertension, as reported by the authors."
Response: We welcome your input and we have given your suggestion the attention their deserve (lines 60-61).
Point 7: "I suggest correcting the definition of resistant hypertension, as reported in the 2018 European Society of Hypertension Guidelines, with "office BP not at goal despite the prescription of three or more classes of antihypertensive medications, including a diuretic, and typically an ACE inhibitor or an Angiotensin receptor blocker and a calcium channel blocker at optimal or best tolerated dose"."
Response: Your suggestions are valued, and we have modified the definition of resistant hypertension (lines 55-57).
Point 8: "I suggest reporting in the main text, more in details, the practical modalities of execution of the intensive lifestyle modification program, that are only mentioned and are unclear."
Response: Please note that we appreciate your suggestions (lines 149-164 and 169-170).
Point 9: "The abstract is too conversational, I suggest mentioning the results of statistical univariate analysis on it."
Response: We have made some revisions to the abstract based on your suggestions (lines 39-43).
Thank you for your feedback.
Round 2
Reviewer 2 Report
Congratulations! I have no more concerns on this paper.